# NAS-Bench-Graph: Benchmarking Graph Neural Architecture Search

**Yijian Qin, Ziwei Zhang, Xin Wang**[*] **Zeyang Zhang, Wenwu Zhu**[*]
Department of Computer Science and Technology
Tsinghua University
Beijing, China 100084
qinyj19@mails.tsinghua.edu.cn, {zwzhang,xin_wang}@tsinghua.edu.cn
zy-zhang20@mails.tsinghua.edu.cn, wwzhu@tsinghua.edu.cn

## Abstract

Graph neural architecture search (GraphNAS) has recently aroused considerable attention in both academia and industry. However, two key challenges seriously hinder the further research of GraphNAS. First, since there is no consensus for the experimental setting, the empirical results in different research papers are often not comparable and even not reproducible, leading to unfair comparisons. Secondly, GraphNAS often needs extensive computations, which makes it highly inefficient and inaccessible to researchers without access to large-scale computation. To solve these challenges, we propose NAS-Bench-Graph, a tailored benchmark that supports unified, reproducible, and efficient evaluations for GraphNAS. Specifically, we construct a unified, expressive yet compact search space, covering 26,206 unique graph neural network (GNN) architectures and propose a principled evaluation protocol. To avoid unnecessary repetitive training, we have trained and evaluated all of these architectures on nine representative graph datasets, recording detailed metrics including train, validation, and test performance in each epoch, the latency, the number of parameters, etc. Based on our proposed benchmark, the performance of GNN architectures can be directly obtained by a look-up table without any further computation, which enables fair, fully reproducible, and efficient comparisons. To demonstrate its usage, we make in-depth analyses of our proposed NAS-Bench-Graph, revealing several interesting findings for GraphNAS. We also showcase how the benchmark can be easily compatible with GraphNAS open libraries such as AutoGL and NNI. To the best of our knowledge, our work is the first benchmark for graph neural architecture search.

## 1 Introduction

With the prevalence of graph data, graph machine learning [77], such as graph neural networks (GNNs) [83, 67], has been widely adopted in diverse tasks, including recommendation systems [66], bioinformatics [55], urban computing [24], physical simulations [54], combinatorial optimization [2], etc. Graph neural architecture search (GraphNAS), aiming to automatically discover the optimal GNN architecture for a given graph dataset and task, is at the front of graph machine learning research and has drawn increasing attention in the past few years [78].

Despite the progress in GraphNAS research, there exist two key challenges that seriously hinder the further development of GraphNAS:

---

[*]Corresponding Authors.

36th Conference on Neural Information Processing Systems (NeurIPS 2022) Track on Datasets and Benchmarks.

- The experimental settings, such as dataset splits, hyper-parameter settings, and evaluation protocols differ greatly from paper to paper. As a result, the experimental results cannot be guaranteed comparable and reproducible, making fair comparisons of different methods extremely difficult.

- GraphNAS often requires extensive computations and therefore is highly inefficient, especially for large-scale graphs. Besides, the computational bottleneck makes GraphNAS research inaccessible to those without abundant computing resources.

Similar challenges have arisen in other domains of NAS research [86], which gives birth to the idea of tabular NAS Benchmarks [72, 12, 27, 75]. Tabular NAS benchmarks provide pre-computed evaluations for all possible architectures in the search space by a table lookup. These benchmarks dramatically boost NAS research, for example, by speeding up the experiments since no architecture training is needed during the search procedure and creating fair comparisons of different NAS algorithms [35]. Inspired by the success of tabular NAS benchmarks and to solve the challenges of GraphNAS, we propose NAS-Bench-Graph[2], the first tabular NAS benchmark that supports unified, reproducible, and efficient evaluations for GraphNAS. Specifically, we first construct a unified GraphNAS search space by formulating the macro space of message-passing as a constrained directed acyclic graph and carefully choose operations from seven GNN candidates. Our proposed search space is expressive yet compact, resulting in 26,206 unique architectures and covering many representative GNNs. We further propose a principled protocol for dataset splits, choosing hyper-parameters, and evaluations. We have trained and evaluated all of these architectures on nine representative graph datasets with different sizes and application domains and recorded detailed metrics during the training and testing process. All the code and evaluation results have been open-sourced. Therefore, our proposed benchmark can enable fair, fully reproducible, and efficient comparisons for different GraphNAS methods.

To explore our proposed NAS-Bench-Graph, we make in-depth analyses from four perspectives with several interesting observations. First, the performance distribution shows that though reasonably effective architectures are common, architectures with extremely good results are rare, and these powerful architectures have diverse efficiencies, as measured by the model latency. Therefore, how to find architectures with both high efficiency and effectiveness is challenging. Second, the architecture distribution suggests that different graph datasets differ greatly in the macro space and operation choices. The cross-datasets correlations further suggest that different graph datasets exhibit complicated patterns and simply transferring the best-performing architectures from similar graph datasets cannot lead to the optimal result. Lastly, detailed architecture explorations demonstrate that architecture space exhibits certain degrees of smoothness, which supports the mutation process in evolutionary search strategies, and deeper parts of architectures are more influential than lower parts, which may inspire more advanced reinforcement learning based search strategies.

To demonstrate the usage of NAS-Bench-Graph, we have integrated it with two representative open libraries: AutoGL [18], the first dedicated library for GraphNAS, and NNI[3], a widely adopted library for general NAS. Experiments demonstrate that NAS-Bench-Graph can be easily compatible with different search strategies including random search, reinforcement learning based methods, and evolutionary algorithms.

Our contributions are summarized as follows.

- We propose NAS-Bench-Graph, a tailored GraphNAS benchmark that enables fair, fully reproducible, and efficient empirical comparisons for GraphNAS research. We are the first to study benchmarking GraphNAS research to the best of our knowledge.

- We have trained and evaluated all GNN architectures in our tailored search space on nine common graph datasets with a unified and principled evaluation protocol. Based on our proposed benchmark, the performance of architectures can be directly obtained without repetitive training.

- We make in-depth analyses for our proposed benchmark and showcase how it can be easily compatible with GraphNAS open libraries such as AutoGL and NNI.

---

[2]https://github.com/THUMNLab/NAS-Bench-Graph
[3]https://github.com/microsoft/nni

## 2 Related Works

### 2.1 Graph Neural Architecture Search

GNNs have shown impressive performance for graph machine learning in the past few years [25, 1, 59, 69, 68, 30, 31, 32, 39, 37, 38]. However, the existing GNNs are manually designed, which require expert knowledge, are labor-intensive, and unadaptable to changes in graph datasets and tasks. Motivated by these problems, automated graph learning has drawn increasing attention in the past few years, including hyper-parameter optimization on graphs [57, 61], and GraphNAS [16, 84, 48, 18, 82, 23, 10, 73, 22, 62, 64, 5, 17]. GraphNAS aims to automatically discover the optimal GNN architectures. Similar to general NAS [14], GraphNAS can be categorized based on the search space, the search strategy, and the performance estimation strategy [78]. For the search space, both micro [16, 80, 34] and macro [63, 15] spaces for the message-passing functions in GNNs are studied, as well as other functions such as pooling [23, 65], heterogeneous graphs [10], and spatial-temporal graphs [45]. Search strategies include reinforcement learning based methods [16, 84, 40], evolutionary algorithms [44, 53, 18], and differentiable methods [82, 79, 81]. Performance estimation strategies can be generally divided into training from scratch [16], hand-designed weight-sharing mechanisms [84, 8], using supernets [48, 6, 18, 49], and prediction-based methods [56]. Despite these progresses, how to properly evaluate and compare GraphNAS methods receives less attention. Our proposed benchmark can support fair and efficient comparisons of different GraphNAS methods as well as motivating new GraphNAS research.

### 2.2 NAS Benchmarks

Since the introduction of NAS-Bench-101 [72], many different benchmarks have been introduced for NAS. However, most previous works focus on computer vision tasks such as NAS-Bench-101 [72], NAS-Bench-201 [12], NATS-Bench [11], NAS-Bench-1shot1 [74], Surr-NAS-Bench [75], HW-NAS-Bench [28], NAS-HPO-Bench-II [20], TransNAS-Bench-101 [13], NAS-Bench-Zero [7], NAS-Bench-x11 [70], and NAS-Bench-360 [58]. Some recent benchmarks also study tabular data (NAS-HPO-Bench [26]), natural language processing (NAS-Bench-NLP [27] and NAS-Bench-x11 [70]), acoustics (NAS-Bench-ASR [41]), and sequence (NAS-Bench-360 [58]). We draw inspiration from these benchmarks and propose the first tailored NAS benchmark for graphs. We provide more comparisons with the existing NAS benchmarks in Appendix D.2.

## 3 Benchmark Design

In this section, we describe our design for the NAS benchmark construction, including the search space (Section 3.1), the datasets used (Section 3.2), and the experiment settings (Section 3.3). We provide some preliminaries of GNNs and GraphNAS in Appendix D.1.

### 3.1 Search Space Design

To balance the effectiveness and efficiency, we design a tractable yet expressive search space. Specifically, we consider the macro search space of GNN architectures as a directed acyclic graph (DAG) to formulate the computation[4], i.e., each computing node indicates a representation of vertices and each edge indicates an operation. Concretely, the DAG contains six nodes (including the input and output node) and we constrain that each intermediate node has only one incoming edge. The resulted DAG has 9 choices, as illustrated in Fig 1. Those intermediate nodes without successor nodes are connected to the output node by concatenation. Besides this macro space, we also consider optional fully-connected pre-process and post-process layers as GraphGym [73] and PasCa [76]. Notice that to avoid exploding the search space, we consider the numbers of pre-process and post-process layers as hyper-parameters, which will be discussed in Section 3.3. Finally, we adopt a task-specific fully connected layer to obtain the model prediction.

For the candidate operations, we consider 7 most widely adopted GNN layers: GCN [25], GAT [59], GraphSAGE [19], GIN [68], ChebNet [9], ARMA [4], and k-GNN [43]. Besides, we also consider Identity to support residual connection and fully connected layer that does not use graph structures.

---

[4]To distinguish the computation graph and graph data, we refer to the computation graph using nodes and edges, and the graph data using vertices and links.

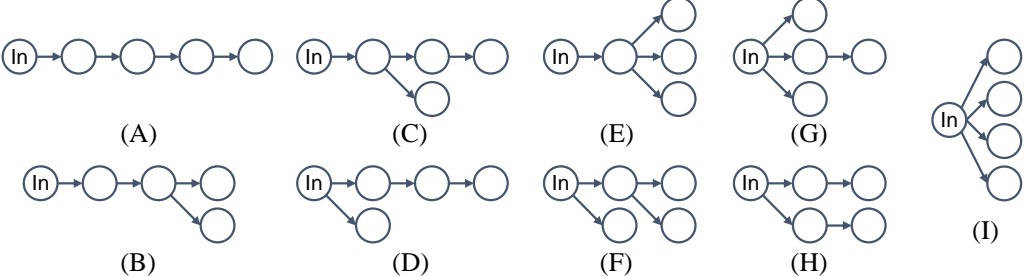

Figure 1: An illustration of the 9 different choices of our macro search space. Each node indicates a representation of vertices and each edge indicates an operation. We omit the output node for clarity.

Table 1: The statistics of the adopted datasets.

| Dataset | #Vertices | #Links | #Features | #Classes | Metric |
|---|---|---|---|---|---|
| Cora | 2,708 | 5,429 | 1,433 | 7 | Accuracy |
| CiteSeer | 3,327 | 4,732 | 3,703 | 6 | Accuracy |
| PubMed | 19,717 | 44,338 | 500 | 3 | Accuracy |
| Coauthor-CS | 18,333 | 81,894 | 6,805 | 15 | Accuracy |
| Coauthor-Physics | 34,493 | 247,962 | 8,415 | 5 | Accuracy |
| Amazon-Photo | 7,487 | 119,043 | 745 | 8 | Accuracy |
| Amazon-Computers | 13,381 | 245,778 | 767 | 10 | Accuracy |
| ogbn-arxiv | 169,343 | 1,166,243 | 128 | 40 | Accuracy |
| ogbn-proteins | 132,534 | 39,561,252 | 8 | 112 | ROC-AUC |

In summary, our designed search space contains 26,206 different architectures (after removing isomorphic structures, i.e., architectures that appear differently but have the same functionality, see Appendix A.5), which covers many representative GNN variants, including methods mentioned above as well as more advanced architectures such as JK-Net [69], residual- and dense-like GNNs [29].

## 3.2 Datasets

We adopt **nine** publicly available datasets commonly used in GraphNAS: Cora, CiteSeer, and PubMed [51], Coauthor-CS, Coauthor-Physics, Amazon-Photo, and Amazon-Computer [52], ogbn-arXiv and ogbn-proteins [21]. The statistics and evaluation metrics of the datasets are summarized in Table 1. These datasets cover different sizes from thousands of vertices and links to millions of links, and various application domains including citation graphs, e-commerce graphs, and protein graphs. More details about the datasets are provided in Appendix A.1.

The dataset splits are as follows. For Cora, CiteSeer, and PubMed, we use the public semi-supervised setting by [71], i.e., 20 nodes per class for training and 500 nodes for validation. For two Amazon and two Coauthor datasets, we follow [52] and randomly split the train/validation/test set in a semi-supervised setting, i.e., 20 nodes per class for training, 30 nodes per class for validation, and the rest for testing. For ogbn-arXiv and ogbn-proteins, we follow the official splits of the dataset.

For ogbn-proteins, we find through preliminary studies that using GIN and k-GNN operations consistently makes the model parameters converge to explosion and therefore results in meaningless results. Besides, GAT and ChebNet will result in out-of-memory errors for our largest GPUs with 32GB of memories. Therefore, to avoid wasting computational resources, we restrict the candidate operations for ogbn-proteins to be GCN, ARMA, GraphSAGE, Identity and fully connected layer. After such changes, there are 2,021 feasible architectures for ogbn-proteins.

## 3.3 Experimental Setting

**Hyper-parameters** To ensure fair and reproducible comparisons, we propose a unified evaluation protocol. Specifically, we consider the following hyper-parameters with tailored ranges:

- Number of pre-process layers: 0 or 1.
- Number of post-process layers: 0 or 1.
- Dimensionality of hidden units: 64, 128, or 256.
- Dropout rate: 0.0, 0.1, 0.2, 0.3, 0.4, 0.5, 0.6, 0.7, 0.8.
- Optimizer: SGD or Adam.
- Learning Rate (LR): 0.5, 0.2, 0.1, 0.05, 0.02, 0.01, 0.005, 0.002, 0.001.
- Weight Decay: 0 or 0.0005.
- Number of training epochs: 200, 300, 400, 500.

For each dataset, we fix the hyper-parameters for all architectures to ensure a fair comparison. Notice that jointly enumerating architectures and hyper-parameters will result in billions of architecture hyper-parameter pairs and is infeasible in practice. Therefore, we first optimize the hyper-parameters to a proper value which can accommodate different GNN architectures, and then focus on the GNN architectures. Specifically, we adopt 30 GNN architectures from our search space as "anchors" and adopt random search for hyper-parameter optimization [3]. The 30 anchor architectures are composed of 20 randomly selected architectures from our search space and 10 classic GNN architectures including GCN, GAT, GIN, GraphSAGE, and ARMA with 2 and 3 layers. We optimize the hyper-parameters by maximizing the average performance of the anchor architectures. The detailed selected hyper-parameters for each dataset are shown in the Appendix A.2.

**Metrics**   During training each architecture, we record the following metric covering both model effectiveness and efficiency: train/validation/test loss value and evaluation metric at each epoch, the model latency, and the number of parameters. The hardware and software configurations are provided in Appendix A.3. Besides, all experiments are repeated three times with different random seeds. The total time cost of creating our benchmark is approximately 8,000 GPU hours (see Appendix A.4).

## 4   Analyses

In this section, we carry out empirical analyses to gain insights for our proposed benchmark. All the following analyses are based on the average performances of three random seeds.

### 4.1   Performance Distribution

We first visualize the distribution of performances, including the accuracy, the latency, and the numbers of parameters, of all architectures in Figure 2. We make several interesting observations. For the effectiveness aspect, many architectures can obtain a reasonably good result, but architectures with exceptionally strong results are still rare. On the other hand, the latency and the numbers of parameters of architectures differ greatly. Since both model effectiveness and efficiency are critical for GraphNAS, we mark the Pareto-optimal with respect to accuracy and latency in the figure. The results show that, even for top-ranking architectures with similar accuracy, their latency varies greatly. In addition, we observe that the number of parameters and the latency are positively correlated in general (details are shown in the Appendix B.1).

### 4.2   Architecture Distribution

As introduced in Section 3.1, our proposed search space mainly consists of the macro space (i.e., the DAG) and candidate GNN operations. To gain insights of how different macro space and GNN operation choices contribute to the model effectiveness, we select the top 5% architectures on each dataset and plot the frequency of the macro search space and operation choices. The results are shown in Figure 3. We make the following observations.

First, there exist significant differences in the macro space choices for different datasets, indicating that different GNN architectures suit different graph data. For example, Cora, CiteSeer and PubMed tend to select a 2-layer DAG, i.e., (E), (F), (G), and (H) (please refer to Figure 1 for the detailed DAGs). PudMed and CS also prefer the 1-layer DAG (I), which is hardly selected in other datasets. Physics, Photo, and Computers show more balanced distributions on the macro space. ogbn-arXiv

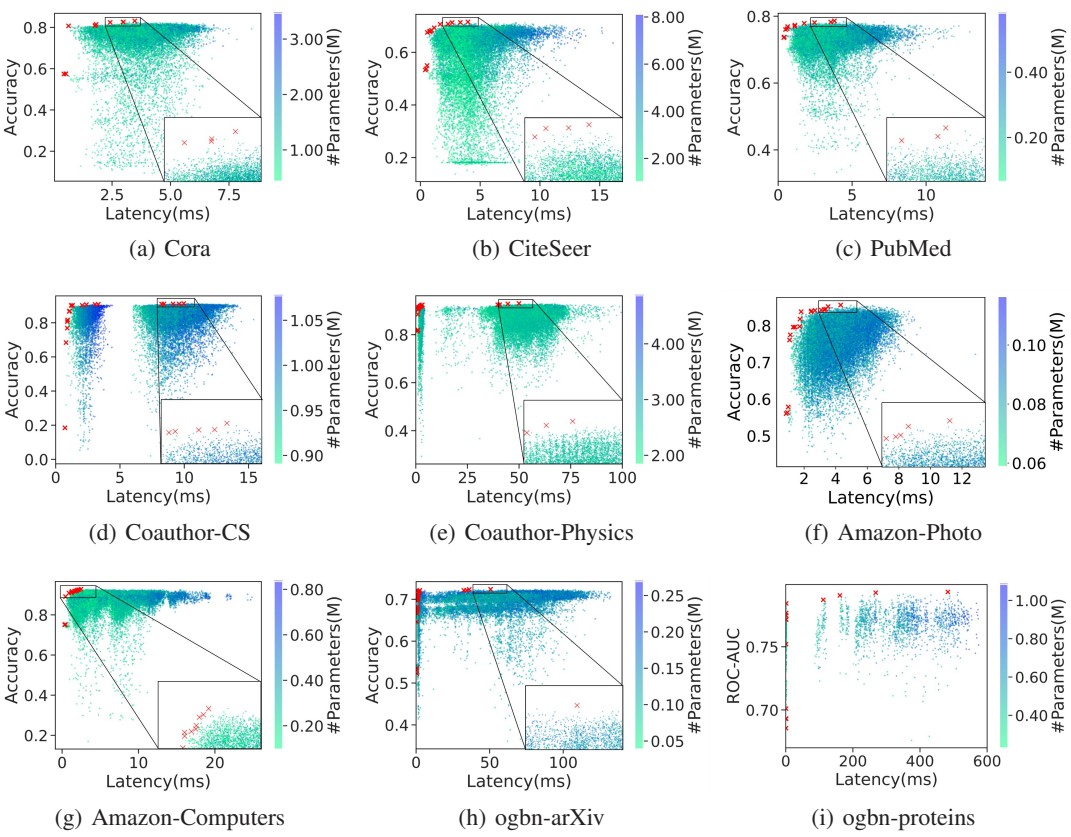

Figure 2: The distribution of accuracy, latency, and the numbers of parameters of all architectures. The Pareto-optimal architectures w.r.t. accuracy and latency are marked with red crosses.

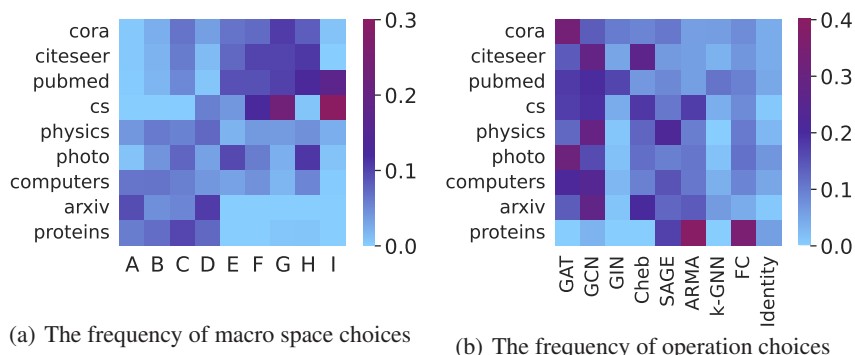

(a) The frequency of macro space choices

(b) The frequency of operation choices

Figure 3: The frequency of the macro space and operation choices in the top 5% architectures of different datasets. Please refer to Figure 1 for the macro space choices.

and ogbn-proteins select deeper architectures more frequently, e.g., the 4-layer DAG (A) and the 3-layer DAG (B), (C), and (D).

As for the operation distribution, different datasets show more similar patterns. GCN and GAT are selected most frequently in almost all datasets. Surprisingly, even though GIN and k-GNN are shown to be theoretically more expressive in terms of the Weifeiler-Lehman test, they are only selected in the relatively small datasets, i.e., Cora, CiteSeer, and PubMed. A plausible reason is that GIN and k-GNN adopt the summation aggregation function in the GNN layers, which is not suitable for node-level tasks in large-scale graphs. Moreover, different from NAS in computer vision where

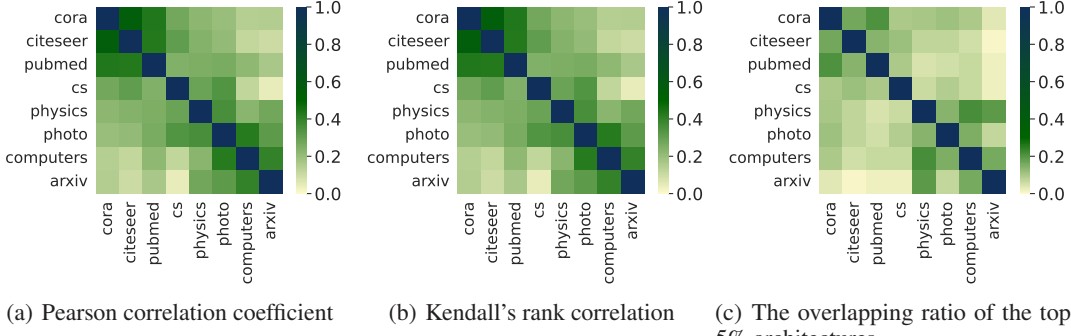

(a) Pearson correlation coefficient

(b) Kendall's rank correlation

(c) The overlapping ratio of the top 5% architectures

Figure 4: The architecture performance correlation across different datasets using three metrics.

Identity operation plays an important role in improving model performance [60], Identity is hardly chosen in any graph datasets. More analyses about the frequency of operations with different depths can be found in the Appendix B.2.

### 4.3 Cross-datasets Correlations

To measure how architectures perform across different datasets, we calculate the performance correlation of all architectures on dataset pairs as You et al. [73]. Specifically, we adopt three metrics: Pearson correlation coefficient, Kendall rank correlation, and the overlapping ratio of the top 5% architectures, i.e., if there are $N$ architectures belonging to the top 5% of both two different datasets in terms of accuracy, then the overlapping ratio is $N/(26, 206 \times 5\%)$. We show the results in Figure 4.

We can observe that the correlation matrix has roughly block structures, indicating that there exist groups of datasets in which architectures share more correlations. For example, Cora, CiteSeer and PubMed generally show strong correlations. The correlations are also relatively high between Physics, Photo, Computers, and ogbn-arXiv. Notice that even for these datasets, only the Pearson correlation coefficient and Kendall rank correlation have large values, while the overlapping ratio of the top 5% architectures is considerably lower (e.g., no larger than 0.3). Since we usually aim to discover best architectures, the results indicate that best-performing architectures in different graph datasets exhibit complicated patterns, and directly transferring the best architecture from a similar dataset as You et al. [73] may not lead to the optimal result. More analyses about the transferability of the optimal architectures on different datasets can be found in the Appendix B.3.

### 4.4 Detailed Explorations in Architectures

Since enumerating all possible architectures to find the best-performing one is infeasible in practice, NAS search strategies inevitably need prior assumptions on the architectures. These explicit or implicit assumptions largely determine the effectiveness of the NAS methods. Next, we explore the architectures in details to verify some common assumptions.

**Evolutionary Algorithm** is one of the earliest adopted optimization methods for NAS [50]. In the mutation process of evolutionary algorithms, i.e., randomly changing choices in the search space, a common assumption is that similar architectures have relatively similar performance so that smoothly mutating architectures is feasible [84]. To verify this assumption, we calculate the performance difference between architectures with different number of mutations together with the average performance difference between randomly chosen architectures as a reference line. The results are shown in Figure 5. We can observe that the performance difference between mutated architectures is considerably smaller than two random architecture, verifying the smoothness assumption in mutations. Besides, we observe that the performance difference increases as the number of mutations in general and changing operation choices usually leads to smaller differences than changing the macro space choices. These observations may inspire further research of evolution algorithms for GraphNAS.

**Reinforcement Learning** (RL) is also widely adopted in NAS [87, 47, 85]. In RL-based NAS, architectures are usually generated by a Markov Decision Process, i.e., deciding the architectures

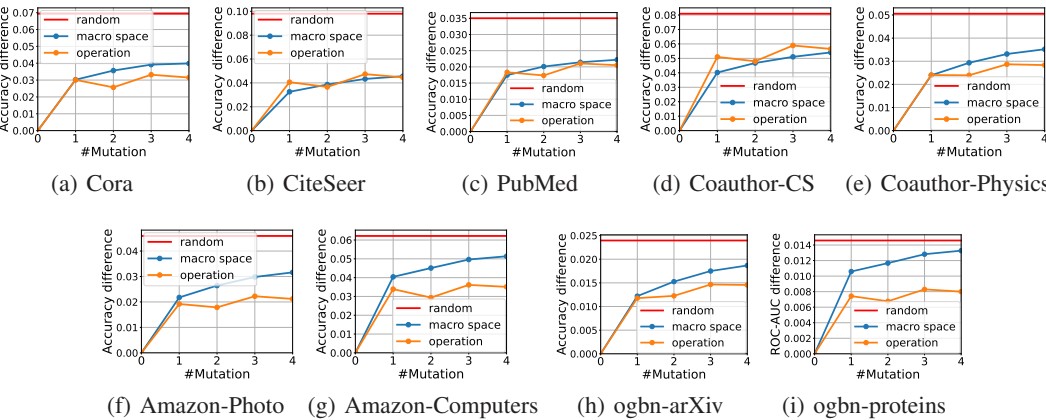

Figure 5: The performance difference between architectures with different number of mutations. The red lines indicate the average performance difference between two random architectures.

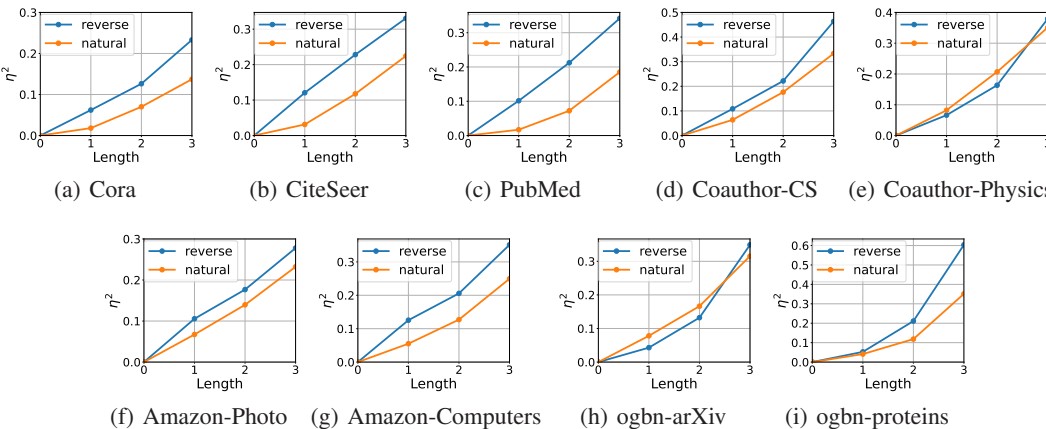

Figure 6: $\eta^2$ for the architecture performance with different lengths of prefixes/suffixes as groups. Larger $\eta^2$ means that the groups can better explain the variance and the corresponding architectures choices are more important for the performance.

using a sequential order [87]. Most GraphNAS methods simply assume a natural order, i.e., generating architectures from lower parts to deeper parts [16]. To gain insights for such methods, we analyze the importance of different positions of architectures. Specifically, we group architectures using their prefixes (i.e., lower-parts choices) and their suffixes (i.e., deeper-parts choices), corresponding to generating architectures using the natural order and the reserve order. Then, we adopt the following metric from analysis of variance [46]:

$$\eta^2 = \frac{SSB}{SST} = \frac{\sum_{k=1}^{K} n_k (\mu_k - \overline{\mu})^2}{\sum_{i=1}^{N} (x_i - \overline{\mu})^2},$$

where $SSB$ and $SST$ are the between-group variations and the total variation, $N$ is the number of architectures, $x_i$ is the accuracy of an architecture $i$, $\overline{\mu}$ is the mean accuracy, $K$ is the number of groups, $n_k$ is the number of architectures in the group $k$, and $\mu_k$ is the mean performance of architectures in the group $k$. In short, a larger $\eta^2$ means the groups can better explain the variances in the samples so that the factors for the group (i.e., architecture choices) are more important. The results of varying the length of the prefixes and suffixed are shown in Figure 6. Somewhat surprisingly, in most datasets, $\eta^2$ in the reverse order is larger than that in the natural order, indicating that deeper parts of the architecture are more influential for the performance than the lower parts. With the length of prefixes and suffixes growing, $\eta^2$ grows significantly. The results indicate that the natural order

Table 2: The performance of NAS methods in AutoGL and NNI using NAS-Bench-Graph. The best performance for each dataset is marked in bold. We also show the performance of the top 5% architecture (i.e., 20-quantiles) as a reference line. The results are averaged over five experiments with different random seeds and the standard errors are shown in the bottom right.

| Library | Method | Cora | CiteSeer | PubMed | CS | Physics | Photo | Computers | arXiv | proteins |
|---------|--------|------|----------|--------|------|---------|-------|-----------|-------|----------|
| AutoGL | GNAS | $82.04_{0.17}$ | $\mathbf{70.89}_{0.16}$ | $77.79_{0.02}$ | $90.97_{0.06}$ | $92.43_{0.04}$ | $92.43_{0.03}$ | $84.74_{0.20}$ | $72.00_{0.02}$ | $\mathbf{78.71}_{0.11}$ |
| | Auto-GNN | $81.80_{0.00}$ | $70.76_{0.12}$ | $77.69_{0.16}$ | $\mathbf{91.04}_{0.04}$ | $92.42_{0.16}$ | $92.38_{0.01}$ | $84.53_{0.14}$ | $\mathbf{72.13}_{0.03}$ | $78.54_{0.30}$ |
| NNI | Random | $82.09_{0.08}$ | $70.49_{0.08}$ | $77.91_{0.07}$ | $90.93_{0.07}$ | $92.35_{0.05}$ | $\mathbf{92.44}_{0.02}$ | $84.78_{0.14}$ | $72.04_{0.05}$ | $78.32_{0.14}$ |
| | EA | $81.85_{0.20}$ | $70.48_{0.12}$ | $\mathbf{77.96}_{0.12}$ | $90.60_{0.07}$ | $92.22_{0.08}$ | $92.43_{0.02}$ | $84.29_{0.29}$ | $71.91_{0.06}$ | $77.93_{0.21}$ |
| | RL | $\mathbf{82.27}_{0.21}$ | $70.66_{0.12}$ | $\mathbf{77.96}_{0.09}$ | $90.98_{0.01}$ | $\mathbf{92.48}_{0.03}$ | $92.42_{0.06}$ | $\mathbf{84.90}_{0.19}$ | $\mathbf{72.13}_{0.05}$ | $78.52_{0.18}$ |
| The top 5% | | 80.63 | 69.07 | 76.60 | 90.01 | 91.67 | 91.57 | 82.77 | 71.69 | 78.37 |

to convert architectures into sequences may not be the optimal solution for GraphNAS and more research could be explored in this direction.

**Differentiable Method.** DARTS [36] in one of the most popular differentiable NAS algorithms. We adopt DARTS on our proposed search space as the search algorithm, and we plot the average weight of different operations Figure 7. Compared to the weights distribution of the top 5% architectures shown in Figure 3(b), we can find that in some datasets, DARTS dispatches the largest weight to the most frequent operations in best-performing architecture, e.g., GAT for Cora and GCN for Amazon-Computers, indicating the effectiveness of DARTS in searching best-performing architectures.

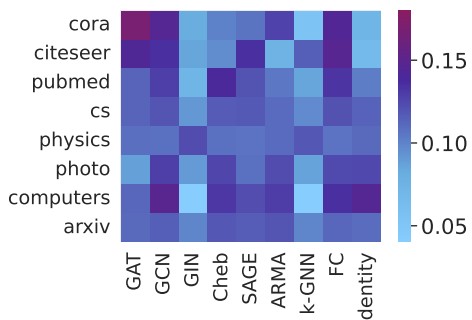

Figure 7: The average weight of different operations obtained using DARTS as the search algorithm

## 5 Example Usages

In this section, we showcase the usage of NAS-Bench-Graph with existing open libraries including AutoGL [18] and NNI [42] (detailed example codes are provided in the Appendix C). Specifically, we run two NAS algorithms through AutoGL: GNAS [16] and Auto-GNN [84]. For NNI, we adopt Random Search [33], Evolutionary Algorithm (EA), and Policy-based Reinforcement Learning (RL). To ensure fair comparisons, we only let each algorithm access the performances of 2% of the all architectures in the search space. We report the results in Table 2. We also show the performances of the top 5% architecture, i.e, the 20-quantiles of each dataset in the table.

From the results, we can observe that all algorithms outperform the top 5% performance, indicating that they can learn informative patterns in NAS-Bench-Graph. However, no algorithm can consistently win on all datasets. Surprisingly, Random Search is still a strong baseline when compared with other methods and even performs the best on two datasets, partially corroborating the findings in [33] for general NAS. The results indicate that further research on GraphNAS is still urgently needed.

To investigate the learning of different GraphNAS methods, we plot the curves of the optimal performance with respect to the number of architectures. The results are shown in Figure 8. We can find that different algorithms behave differently. For example, EA and AGNN show a few "jumps", i.e., the performance largely increases, while RL shows more smooth increasing patterns. Taking closer looks at the learning curve may provide inspirations for developing new algorithms for GraphNAS.

## 6 Conclusion and Future Work

In this paper, we propose NAS-Bench-Graph, the first tailored NAS benchmark for graph neural networks. We have trained all 26,206 GNN architectures in our designed search space on nine representative graph datasets with a unified evaluation protocol. NAS-Bench-Graph can support

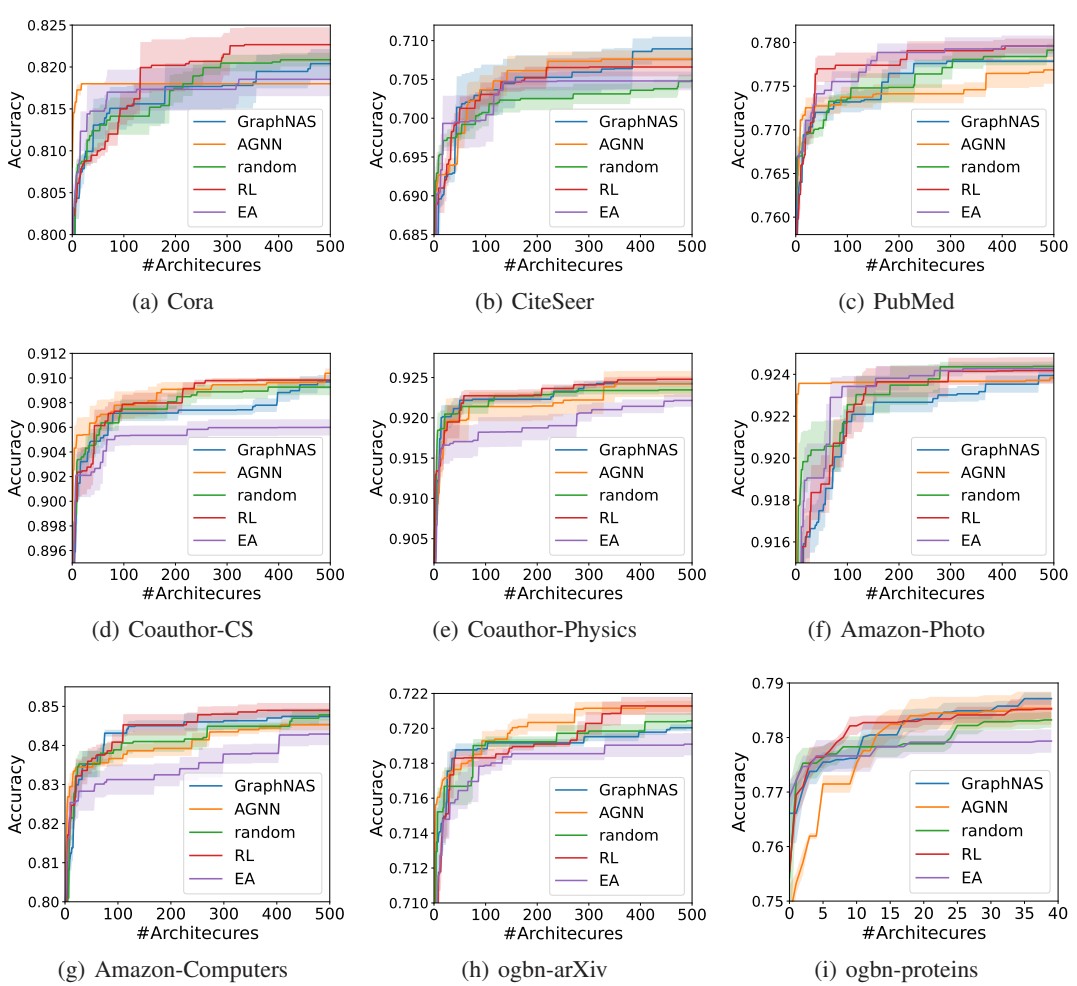

Figure 8: The learning curve of the optimal performance with respect to the number of searched architectures. All results are averaged over five experiments with different random seeds. The standard errors are shown in the background.

unified, reproducible, and efficient evaluations for GraphNAS. We also provide in-depth analyses for NAS-Bench-Graph and show how it can be easily compatible with the existing GraphNAS libraries.

Since constructing tabular NAS benchmarks consumes extensive computational resources, our proposed NAS-Bench-Graph has a relatively limited search space (i.e., 26,206 unique GNN architectures). A possible direction is constructing surrogate benchmarks for GraphNAS to allow larger search spaces. We also plan to extend our proposed benchmark to other graph tasks besides node classification, such as link prediction and graph classification.

## Acknowledgments and Disclosure of Funding

This work was supported in part by the National Key Research and Development Program of China No. 2020AAA0106300, National Natural Science Foundation of China (No. 62250008, 62222209, 62102222), China National Postdoctoral Program for Innovative Talents No. BX20220185, and China Postdoctoral Science Foundation No. 2022M711813.

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
