# OpenReview forum: "NAS-Bench-Graph: Benchmarking Graph Neural Architecture Search"
_NeurIPS.cc/2022/Track/Datasets_and_Benchmarks — NeurIPS 2022 Datasets and Benchmarks _

### Official Review · Reviewer_LkTe · 2022-07-12
**Good analysis and codebase, but needs benchmark more existing methods.**

**Rating:** 5
**Confidence:** 4
**Clarity:** The paper is well written and easy to…

**Strengths:**

+ The motivation of this benchmark is good, which calls for a unified and fair benchmark for GNN NAS algorithms.

+ The proposed benchmark is first presented in GNN NAS domain, to the reviewer’s best knowledge.

+ The visualization Fig. 2 and 4 provide good insights to analyze the latency-performance trade-off and cross-dataset architecture transferability.


**Weaknesses:**

- Most of the experiments are done on their proposed search space, which may vary across different methods. This seems to be a mismatch with the goal of a benchmark, which is often designed to compare different methods under a fair experimental setting.
- The chosen datasets are not diverse enought. Only homophily data are considered. It would be better to consider heterophily data.
- The benchmark paper does not compare with enough baseline methods, such as SANE, GASSO, AutoCoG.


**Additional Feedback:**

1. The macro search space presents some multi-branch architectures. How do authors combine their outputs?
2. This paper presents random, EA and RL based methods. Why differential architecture search based methods are not presented.

The major limitation of this paper is the insufficient compared baselines. Most of the experiments and discussion are done on their proposed search space and methods. As a benchmark paper, the reviewer may suggest including the comparison with more existing methods. Also, it is recommended to involve more heterophilic datasets.

**Correctness:**

As far as the reviewer understood, the claims are correct though the goal of a benchmark somewhat mismatches with what this paper presents.

**Documentation:**

The paper uses public dataset and the code is released.

**Ethics:**

No ethical concern yet.

**Relation To Prior Work:**

This paper is the first GNN NAS benchmark as far as the reviewer knows. But some related baselines is missing.

**Summary And Contributions:**

The paper presents a NAS for GNN benchmark, that is fair, comprehensive, and reproducible. The authors provide a framework for GNN NAS unifying the codebase hyperparameters, datasets, and evaluation protocols. They propose a new search space and conduct analysis and visualization on the searched architectures and their transferability.

---

> ### Author Response · Authors · 2022-08-13
> **Response to Reviewer LkTe**
>
> We thank the reviewer for the reviewing efforts and constructive comments. We address your concerns point by point.
>
> **Comment 1: Most of the experiments are done on their proposed search space, which may vary across different methods. This seems to be a mismatch with the goal of a benchmark, which is often designed to compare different methods under a fair experimental setting.**
>
> A1: Thank you for your comment. Following the literature on NAS benchmarks in other domains (e.g., NAS-Bench-101/201/301, etc.), our proposed NAS-Bench-Graph has a unified search space for different datasets and focuses on comparing the search strategies instead of directly comparing the results of existing methods. Considering the literature and that we have adopted the same evaluation protocol (e.g., hyper-parameters, evaluation metrics, etc.), we believe our proposed benchmark is fair in the comparison.
>
> **Comment 2: The chosen datasets are not diverse enough. Only homophily data are considered. It would be better to consider heterophily data.**
>
> A2: Thank you for the suggestion. We agree that heterophily graphs are important and novel directions for GNNs. Since we are the first to study GraphNAS benchmark, we mainly focus on homophily graphs, and leave extensions to heterophily graphs as important future works.
>
> **Comment 3: The benchmark paper does not compare with enough baseline methods, such as SANE, GASSO, AutoCoG.**
>
> A3: Thanks for the comment. Our proposed benchmark mainly aims to provide a fair comparison for search strategies with a unified search space, following the common literature of NAS Benchmarks. Since these methods you mentioned focus on other aspects of GraphNAS, e.g., GASSO and AutoCoG focus on structure learning and SANE proposes a new search space, these methods are not suitable to be compared in our benchmark.
>
> **Comment 4: The macro search space presents some multi-branch architectures. How do authors combine their outputs?**
>
> A4: Thanks for your comment. We concatenate the intermediate nodes (i.e., multi-branch architectures) into the output (L112, P3).
>
> **Comment 5: This paper presents random, EA and RL based methods. Why differential architecture search based methods are not presented.**
>
> A5: Thank you for the comment. Following your suggestion, we have added analysis of differentiable NAS algorithms in Appendix B.4 in the revised version. Briefly speaking, we find that the weights of operations in DARTS are consistent with the frequencies of top architectures in some cases.

---

### Official Review · Reviewer_zW3u · 2022-07-25
**An interesting and useful benchmark for graph neural architecture search**

**Rating:** 6
**Confidence:** 3
**Correctness:** The claims made in this paper are rea…
**Clarity:** This paper is well written and easy t…

**Strengths:**

+Clear GraphNAS search space abstraction and benchmark construction procedure.
+Detailed and intuitive analysis of the architecture space.


**Weaknesses:**

The authors said NAS-Bench-Graph contains 26,206 different architectures, and a detailed explanation of how the number is derived would be helpful.

Typo in line 30: “...key challenges that seriously hind the”, hind->hinder.


**Additional Feedback:**

N/A

**Documentation:**

This paper has sufficient details on how the benchmark is built. Also the data and user guide of GraphNAS benchmark is publicly available via the Github link.

**Ethics:**

No issue identified.

**Relation To Prior Work:**

I am satisfied with the discussion on the relation to prior work.

**Summary And Contributions:**

To address two key challenges in GraphNAS 1) unified and reproducible experiments 2) excessive computation requirements. The authors propose NAS-Bench-Graph, a benchmark for graph neural architecture search (GraphNAS) including data points from 26K GNN architectures on nine prevalent graph datasets. In addition, the authors conducted detailed analysis such as the visualized distribution of accuracy, latency, the number of parameters, and the compatibility of NAS-Bench-Graph to existing NAS algorithms.  The authors also demonstrate the effectiveness of NAS-Bench-Graph by searching GNN architecture with open-sourced libraries on NAS-Bench-Graph and report the performance of the searched network.

---

> ### Author Response · Authors · 2022-08-13
> **Response to Reviewer zW3u**
>
> We thank the reviewer for the reviewing efforts and constructive comments. We address your concerns point by point.
>
> **Comment 1: The authors said NAS-Bench-Graph contains 26,206 different architectures, and a detailed explanation of how the number is derived would be helpful.**
>
> A1: Thank you for your comment. Following your suggestion, we have added our method for removing isomorphic architectures in Appendix A.5 in the revised version. In our proposed search space, we have 9 candidate operations for 4 choices in the marco space, so there are $9^4$ operation choices for each macro space architecture. Since we have 9 types of macro spaces, we have $9^4*9=59,049$ architectures in total. However, some of these architectures are isomorphic, e.g., an identity operation followed by a GCN layer is equivalent to a GCN layer followed by an identity operation. We obtain 26,206 unique architectures after removing these duplicates.
>
> **Comment 2: Typo in line 30: “...key challenges that seriously hind the”, hind->hinder.**
>
> A2: Thank you for your suggestion. We have fixed the typo in the revised version.

---

### Official Review · Reviewer_UeLP · 2022-07-26
**Accept with some more room for improvement in the rebuttal**

**Rating:** 8
**Confidence:** 4
**Clarity:** 1. The paper is clear in its writing,…

**Strengths:**

1. **Impact and relevance to NeurIPS community**: NAS researchers in general will find the benchmark useful and build on it. NAS for GNNs in particular has seen several works in NeurIPS and other top conferences, so NAS-Bench-Graph is highly relevant to the NeurIPS community.
2. **Originality**: NAS-Bench-Graph follows many previous tabular benchmarks for NAS, it is, however, the first such benchmark for NAS for GNNs and the first NAS benchmark for the graph modality. The benchmark features a macro and micro component for the search space, which not many benchmarks do.
3. **Accessibility and accountability**: The main benchmark data is hosted on github, with the complete data available on Figshare with its maintenance guarantees etc. The authors provide code to read the data in the common data format of a python dictionary. Further, some documentation is provided in the form of a README that shows the basic usage of the benchmark. License for code and data is given. I did not try running the code.
4. **Miscellaneous**: NAS-Bench-Graph features a large number of (varied) datasets (9) compared to many other NAS benchmarks, which allows for more comprehensive benchmarking. Performance metrics are recorded for all epochs, allowing, e.g., also multi-fidelity algorithms to be benchmarked with NAS-Bench-Graph. The paper includes several insightful analysis that provide a good intuition of the benchmark characteristics and, e.g., add evidence for the surprisingly strong performance of random search. The authors include a detailed explanation for the systematic setting of hyperparameters.

**Weaknesses:**

1. **Accessibility and accountability**: The code that was used to create the benchmark data is not open sourced. The same goes for the experiments and empirical analysis. The python dependencies for the benchmark are not specified. Ideally, NAS-Bench-Graph could be installed via pip from github or pypi. Installation instructions in general are not provided.
2. **Empirical protocol**: The empirical evaluation does not quantify meassurement error, e.g., by providing error bounds, making it hard to assess how statistically sound their analysis is. This is acknowledged in the checklist which I appreciate. Further, the empirical evaluation only concerns final performance after ~200 evaluations and not full optimization curves.

**Additional Feedback:**

1. A discussion on potential negative societal impacts was crossed as yes in the checklist, but I did not find it.
2. Can make github links clickable in paper
3. Do not say "we do x as in [11] but "we do x as xy et al. [11]"
4. The choice of HPs involves optimizing for random samples and common architectures, which introduces a bias. This is a minor point and HPs have to be chosen somehow.
5. How much compute resources did go into creating this benchmark and how costly is it to run a training from scratch for a typical architecture?
6. Figure 5 and 6 are not legible without zooming in, please fix this. Ideally the font should be the same size as the main text.

**Correctness:**

* The claims made in the submission are correct and sound.
* The experiment design is lacking in some aspects as described under weaknesses.

**Documentation:**

See comments on "accessibility and accountability" in strengths / weaknesses.

**Ethics:**

No ethical concerns.

**Relation To Prior Work:**

1. Related work on NAS for GNNs is covered well.
2. A more comprehensive comparison to existing NAS benchmarks, e.g., in the form of a table in the Appendix would help, but the main discussion on how this work differs from previous contributions in its data modality is sufficient.
3. The paper does not include a discussion on the drawbacks and positives of tabular NAS benchmarks compared to surrogate benchmarks.

**Summary And Contributions:**

**Edit 08-25**: I increase my score from 7 to 8 as the authors addressed all my concerns and acted upon my suggestions for improvements. I  am also happy about several improvements in response to other reviewers / the AC, especially including a discussion on limitations in the main text. Concerns raised from other reviewers are either not convincing to me or adequately addressed by the authors.

---

The paper presents a tabular NAS benchmark for GNN architectures spanning nine graph datasets. The tabular data includes several metrics and will allow future research to study NAS for GNNs in an efficient and principled fashion. The authors also use their benchmark data to derive insights such as different graph datasets requiring different architectural choices for best performance.

---

> ### Author Response · Authors · 2022-08-13
> **Response to Reviewer UeLP**
>
> We thank the reviewer for the reviewing efforts and constructive comments. We address your concerns point by point.
>
> **Comment1: Accessibility and accountability: The code that was used to create the benchmark data is not open sourced. The same goes for the experiments and empirical analysis. The python dependencies for the benchmark are not specified. Ideally, NAS-Bench-Graph could be installed via pip from github or pypi. Installation instructions in general are not provided.**
>
> A1: Thanks for your detailed comments. Following your suggestion, we have provided the code for creating the benchmark and analysis in the supplementary, and we have updated the python dependencies and PyPI installation instructions in the Github repository.
>
> **Comment2: Empirical protocol: The empirical evaluation does not quantify measurement error, e.g., by providing error bounds, making it hard to assess how statistically sound their analysis is. This is acknowledged in the checklist which I appreciate. Further, the empirical evaluation only concerns final performance after ~200 evaluations and not full optimization curves.**
>
> A2: Thank you for your comment. Following your suggestion, we have added error bounds in the table and provided the learning curves in Appendix B.5 in the revised version.
>
> **Comment3: Some background on GNNs is missing, as the benchmark is relevant to NAS researchers in general and not just researchers familiar with GNNs.**
>
> A3: Thank you for your comment. Following your suggestion, we have added some preliminaries for GNNs in Appendix D.1 in the revised version.
>
> **Comment4: A more comprehensive comparison to existing NAS benchmarks, e.g., in the form of a table in the Appendix would help, but the main discussion on how this work differs from previous contributions in its data modality is sufficient.**
>
> A4: Thank you for your comment. Following your suggestion, we have added a table to clarify the differences with previous NAS benchmarks in Appendix D.2 in the revised version.
>
> **Comment 5: The paper does not include a discussion on the drawbacks and positives of tabular NAS benchmarks compared to surrogate benchmarks.**
>
> A5: Thank you for your comment. Following your suggestion, we have added some discussions about tabular NAS benchmarks in Appendix D.2 in the revised version.
>
> **Comment 6: A discussion on potential negative societal impacts was crossed as yes in the checklist, but I did not find it.**
>
> A6: Thank you for your comment. We have briefly discussed this issue in Appendix E Broader Impact and Discussions.
>
> **Comment 7: Can make github links clickable in paper.**
>
> A7: Thank you for your comment. We have fixed the link in the revised version.
>
> **Comment 8: Do not say "we do x as in [11] but "we do x as xy et al. [11]"**
>
> A8: Thank you for your comment. We have fixed the expression in the revised version.
>
> **Comment 9: The choice of HPs involves optimizing for random samples and common architectures, which introduces a bias. This is a minor point and HPs have to be chosen somehow.**
>
> A9:  Thank you for your insightful comment. Since enumerating hyper-parameter and architecture pairs will result in unbearable computational costs, we propose a straightforward and heuristic approach to choosing the hyper-parameters. We leave further studying how to choose hyper-parameters as future works.
>
> **Comment 10: How much compute resources did go into creating this benchmark and how costly is it to run a training from scratch for a typical architecture?**
>
> A10: Thank you for your suggestion. Following your comment, we have provided the consumed computation resource in Appendix A.4. In short, the creation of our benchmark costs approximately 8,000 GPU hours.
>
> **Comment 11: Figure 5 and 6 are not legible without zooming in, please fix this. Ideally the font should be the same size as the main text.**
>
> A11: Thank you for your comment. Following your suggestion, we have revised Figure 5 and Figure 6 in the revision.

---

> > ### Comment · Reviewer_UeLP · 2022-08-17
> > **All concerns adressed. Some additional comments.**
> >
> > Thanks for addressing all my concerns and making several improvements to your submission. I have some additional comments below.
> >
> > > A2: Thank you for your comment. Following your suggestion, we have added error bounds in the table and provided the learning curves in Appendix B.5 in the revised version.
> >
> > What do the error bounds represent? It would be best if you state that in the caption of the table and also how many seeds you used.
> >
> > Having learning curves for all datasets would be good and also having errors bounds. Stating how many seeds you used in the caption would help, too.
> >
> >
> > > A11: Thank you for your comment. Following your suggestion, we have revised Figure 5 and Figure 6 in the revision.
> >
> > The figures are much easier to read now, thanks! The axes and legends are still very tiny on a print out though (for Figure 2 - 4, too).

---

> > > ### Author Response · Authors · 2022-08-25
> > > **Additional response to Reviewer UeLP**
> > >
> > > We thank the reviewer for the response and additional comments. We further revise our paper as follows.
> > >
> > > **Comment1: What do the error bounds represent? It would be best if you state that in the caption of the table and also how many seeds you used. Having learning curves for all datasets would be good and also having errors bounds. Stating how many seeds you used in the caption would help, too.**
> > >
> > > Thank you for the suggestion. Following your suggestion, we have added detailed descriptions of the error bounds and random seeds in the figure caption in the revised version. We show learning curves with error bounds for all datasets in the revised Appendix.
> > >
> > > **Comment2: The figures are much easier to read now, thanks! The axes and legends are still very tiny on a print out though (for Figure 2 - 4, too).**
> > >
> > > Thank you for the suggestion. Following your suggestion, we redraw Figure 2~6 in the revised version to make them easier to read.

---

> > > > ### Comment · Reviewer_UeLP · 2022-08-25
> > > > **Increasing score to 8 and some additional suggestions**
> > > >
> > > > I increase my score from 7 to 8 as the authors addressed all my concerns and acted upon my suggestions for improvements. I am also happy about several improvements in response to other reviewers / the AC, especially including a discussion on limitations in the main text. Concerns raised from other reviewers are either not convincing to me or adequately addressed by the authors.
> > > >
> > > > > Thank you for the suggestion. Following your suggestion, we have added detailed descriptions of the error bounds and random seeds in the figure caption in the revised version. We show learning curves with error bounds for all datasets in the revised Appendix.
> > > >
> > > > I would welcome you adding more seeds to your evaluation (currently only 3). It may also be worth it to use standard error instead of standard deviation (to actually quantify measurement error and not population spread), but that is a matter of taste to some extend.

---

> > > > > ### Author Response · Authors · 2022-08-27
> > > > > **Additional response to Reviewer UeLP**
> > > > >
> > > > > We are very grateful for the reviewer's recognition of our work and the increasing of the score.
> > > > >
> > > > > **Comment: I would welcome you adding more seeds to your evaluation (currently only 3). It may also be worth it to use standard error instead of standard deviation (to actually quantify measurement error and not population spread), but that is a matter of taste to some extend.**
> > > > >
> > > > > Thank you for the suggestion. We have increased the number of seeds to 5 and update the results in the revised version. Besides, it makes sense to use the standard error instead of the standard deviation in our case, and we have changed the statistics in the revised version.

---

### Official Review · Reviewer_8asK · 2022-07-27
**ok but not good**

**Rating:** 4
**Confidence:** 4
**Correctness:** Good.

**Strengths:**

The studied problem is novel. The authors tried to benchmark GraphNAS algorithms, which haven’t been studied by anyone else.

The proposed benchmark is compatible with existing GraphNAS libraries such as AutoGL and NNI.


**Weaknesses:**

The studied problem is novel, but not important/interesting. Different NAS algorithms might have different performances on different problems/tasks/datasets. Considering the NAS is usually used to find the best model structure for a specific problem/task/dataset, what are the advantages of benchmarking different NAS?

The author should compare with GraphGym [1], considering this is one of the most important and most common-used GraphNAS libraries, and a lot of designs in this paper are inspired by GraphGym, e.g., the search space.

The authors only conducted experiments on node classification tasks. Link prediction and graph classification are also important tasks in graph learning.

The authors just generally compared the “evolutionary algorithm” and “RL algorithm” in the paper. However, there are a lot of specific GraphNAS baselines that the authors ignored and didn’t compare [2, 3, 4].

Lack of analysis and insights from the experiments. What are the take-aways from benchmarking different NAS algorithms?


[1] You, Jiaxuan, Zhitao Ying, and Jure Leskovec. "Design space for graph neural networks." Advances in Neural Information Processing Systems 33 (2020): 17009-17021.

[2] Gao, Yang, et al. "Graphnas: Graph neural architecture search with reinforcement learning." arXiv preprint arXiv:1904.09981 (2019).

[3] Zhao, Huan, Quanming Yao, and Weiwei Tu. "Search to aggregate neighborhood for graph neural network." arXiv preprint arXiv:2104.06608 (2021).

[4] Shi, Min, et al. "Evolutionary architecture search for graph neural networks." arXiv preprint arXiv:2009.10199 (2020).

[5] Zhou, Kaixiong, et al. "Auto-gnn: Neural architecture search of graph neural networks." arXiv preprint arXiv:1909.03184 (2019).


**Additional Feedback:**

Please see the comments above.

**Clarity:**

Fair. The authors should provide more illustrations and clarifications regarding the motivation and analysis.

**Documentation:**

Good.

**Ethics:**

N/A.

**Relation To Prior Work:**

The authors claimed that they were the first to benchmark GraphNAS algorithms, but didn’t compare with a lot of specific algorithms. Please see the comments above.


**Summary And Contributions:**

The authors propose NAS-Bench-Graph, a tailored benchmark for GraphNAS. The authors trained and evaluated different GNN architectures on nine graph datasets. The authors showed that the proposed benchmark is compatible with existing GraphNAS libraries such as AutoGL and NNI.

---

> ### Author Response · Authors · 2022-08-13
> **Response to Reviewer 8asK**
>
> We thank the reviewer for the reviewing efforts and constructive comments. We address your concerns point by point.
>
> **Comment1: The studied problem is novel, but not important/interesting. Different NAS algorithms might have different performances on different problems/tasks/datasets. Considering the NAS is usually used to find the best model structure for a specific problem/task/dataset, what are the advantages of benchmarking different NAS?**
>
> A1: Thanks for your comment. We would like to clarify the importance of benchmarking GraphNAS as follows. We agree that the ultimate goal of NAS is to find the best architecture for each specific problem/task/dataset. However, during studying and researching GraphNAS, different methods inevitably need to be properly evaluated and compared to consistently make progress, which makes reproducible, fair, and efficient benchmarks extremely important and necessary. Similar efforts have been studied for other domains, e.g., NAS-Bench-101/201/301, etc., for computer vision, which has drawn widespread attention. We are the first to study benchmarking NAS for graphs, which we believe is critical and urgently needed for the community.
>
> **Comment2: The author should compare with GraphGym [1], considering this is one of the most important and most common-used GraphNAS libraries, and a lot of designs in this paper are inspired by GraphGym, e.g., the search space.**
>
> A2: Thanks for your insightful comment. We would like to clarify the difference with GraphGym as follows. GraphGym is a pioneering work studying the design space of GNNs, and we have indeed drawn great inspiration from it in developing our benchmark. **However, GraphGym is a public library and codebase for GNNs but not a NAS-Benchmark with the following two fundemtnal differences**. First, GraphGym **solely focuses on the search space and does not consider the search strategy** of GNN architectures. Second and more importantly, **we have trained and provided the performance of all possible architectures in our search space**, consuming approximately 8,000 GPU hours. Then, the evaluation of GraphNAS can be obtained by look-up tables for extremely efficient comparisons, which cannot be achieved by GraphGym. We have added these discussions in the revised version.
>
> **Comment3: The authors only conducted experiments on node classification tasks. Link prediction and graph classification are also important tasks in graph learning.**
>
> A3: Thank you for the suggestion. Since we are the first to study GraphNAS benchmark, we mainly focus on node classification, the most widely adopted task for GNNs. We agree that more tasks, such as link prediction and graph classification, could further improve our paper, and leave these extensions as important future works.
>
> **Comment4: The authors just generally compared the “evolutionary algorithm” and “RL algorithm” in the paper. However, there are a lot of specific GraphNAS baselines that the authors ignored and didn’t compare.**
>
> A4: Thank you for the comment. In Table 2, we also showcase how to use our proposed benchmark with two specific GraphNAS baselines: GNAS and Auto-GNN. Notice that our benchmark mainly focuses on providing a fair comparison for search strategies with a unified search space, following the common literature of NAS Benchmarks.

---

> ### Author Response · Authors · 2022-08-13
> **Response to Reviewer 8asK (Part 2)**
>
> **Comment5: Lack of analysis and insights from the experiments. What are the take-aways from benchmarking different NAS algorithms?**
>
> A5: Thanks for the comment. As partially explained in A1, the main goal of NAS-Bench-Graph is to provide a benchmarking for reproducible, fair, and efficient comparisons of different methods so that future GraphNAS research could be easily and properly evaluated, which in turn promotes the direction. Therefore, comparing the existing methods and drawing take-away messages is not the main focus of this paper. Nevertheless, we indeed have several interesting findings while analyzing our proposed benchmark, as shown in Section 4 of the paper. We show some as follows.
> 1. Many architectures can obtain a reasonably good result, but architectures with exceptionally strong results are still rare, and the latency of top-ranking architectures varies greatly.
> 2. Different datasets prefer different marco spaces, especially the number of layers, more than operation choices.
> 3. The architecture performance correlation matrix has roughly block structures, indicating that there exist groups of datasets in which architectures share more correlations.
> 4. The performance difference between mutated architectures is considerably smaller than two random architectures, verifying the smoothness assumption in mutations in the EA algorithm.
> 5. In most datasets, deeper parts of the architecture are more influential for the performance than the lower parts.

---

### Official Review · Reviewer_DnGM · 2022-07-28
**Review for NAS-Bench-Graph**

**Rating:** 7
**Confidence:** 2

**Strengths:**

 - Following a procedure that has worked in other fields is very sound and good practice
 - Can be very useful for many practical purposes

**Weaknesses:**

 - Not completely clear/insisting on the main point of the article which seems to be the benchmarking of different methods, as it seems that there is not much novelty on the different methods of NAS

**Additional Feedback:**

Are the best structures for the similar datasets (from the Cross-Dataset Correlation part) correlated too ?

**Clarity:**

It is well-written, but it feels like it gets a bit lost in technicality and loses a bit of clearness on its main point.

**Correctness:**

The article seems constructed in a sound way, and is extensive on the methods used to create the data for the benchmark

**Documentation:**

The code is available on a github repository and the full detail for training/validating/testing is available on figshare.

**Relation To Prior Work:**

Related works are presented in part 2 quite clearly.

**Summary And Contributions:**

This article presents a benchmark for Neural Architecture Search aiming at giving a baseline protocol to efficiently compare different architecture, in a reproducible manner, and a easily usable database for any practician, mimicking the progresses made in other NAS domains.
It presents its way of searching an efficient architecture using 7 base and typical Graph Neural Network (GNN) layer, and tests them on 9 open datasets, also well-known and relevant in GNN research, using different approaches (namely evolutionary algorithms and reinforcement learning).
Another part presents insights on the distributions for the models, with their performance, how much each was used. Some additional information is provided on the correlation between datasets.

---

> ### Author Response · Authors · 2022-08-13
> **Response to Reviewer DnGM**
>
> We thank the reviewer for the reviewing efforts and constructive comments. We address your concerns point by point.
>
> **Comment1: Not completely clear/insisting on the main point of the article which seems to be the benchmarking of different methods, as it seems that there is not much novelty on the different methods of NAS.**
>
> A1: Thanks for your comment. We would like to clarify the novelty of our paper as follows. The main point of our NAS-Bench-Graph is to **provide reproducible, fair, and efficient evaluations of existing and future GraphNAS methods**.  Using our proposed benchmark, new GraphNAS methods could be more easily and properly evaluated and compared, which can greatly promote the research of GraphNAS. Besides, we make in-depth analyses based on our benchmark (please refer to Section 4 of the paper), which reveals interesting findings about GraphNAS and could inspire novel methods.
>
> **Comment2: Are the best structures for the similar datasets (from the Cross-Dataset Correlation part) correlated too ?**
>
> A2: Thanks for the comment. Following your suggestion, we have added an experiment on the transferability of the best architecture of each dataset to other datasets. The results are provided in Appendix B.3 in the revised version. Briefly speaking, we find that in some cases the best architecture can transfer well.

---

### Official Review · Reviewer_KKFf · 2022-07-29
**An interesting work**

**Rating:** 7
**Confidence:** 5
**Correctness:** Good.
**Clarity:** The presentation of the paper is very…

**Strengths:**

Please check the above contribution part.

**Weaknesses:**

1. The search space is kind of smaller, and the datasets are only for node-level tasks, thus may limit the usage of the benchmark.

2. In Section 4, it will be more interesting if the authors can analyze the influences of differentiable search algorithms beyond RL and EA.

3. Some related works are missing, e.g.,

### Refs
1. Qin et al., Graph Neural Architecture Search Under Distribution Shifts. ICML 2022
2. Wei et al., Designing the topology of graph neural networks: a novel feature fusion perspective. WebConf 2022
3. Cai et al., Rethinking graph neural architecture search from message-passing. CVPR 2021
4. Zhao et al., Search to aggregate neighborhood for graph neural network. ICDE 2021
5. Want et al., AutoGEL: An Automated Graph Neural Network with Explicit Link Information. NeurIPS 2021

**Additional Feedback:**

None

**Documentation:**

Good.

**Relation To Prior Work:**

Yes.

**Summary And Contributions:**

In this paper, the authors propose a benchmark for graph neural architecture search. Based on a tailored search space, they evaluate different search algorithms of existing methods on graph neural architecture search. They further give analyzes of the benchmark in terms of different aspects.

This is the first work to design a benchmark for graph neural architecture search and extensive experiments are conducted and comprehensive analysis is given, which can help us better understand the progress of this topic. Further, it can provide a database of GNN architectures for quick look-up, which can accelerate the following-up research of graph neural architecture.

---

> ### Author Response · Authors · 2022-08-13
> **Response to Reviewer KKFf**
>
> We thank the reviewer for the reviewing efforts and constructive comments. We address your concerns point by point.
>
> **Comment1: The search space is kind of smaller, and the datasets are only for node-level tasks, thus may limit the usage of the benchmark.**
>
> A1: Thank you for the suggestion. Since we are the first to study GraphNAS benchmark, we mainly focus on node classification, the most widely adopted task for GNNs, and a representative search space covering many representative GNNs. We agree that more tasks and a larger search space could further improve our paper, and leave these extensions as important future works.
>
> **Comment2: In Section 4, it will be more interesting if the authors can analyze the influences of differentiable search algorithms beyond RL and EA.**
>
> A2: Thank you for the comment. Following your suggestion, we have added analysis of differentiable NAS algorithms in Appendix B.4 in the revised version. We find that the weights of operations in DARTS are consistent with the frequencies of top architectures in some cases.
>
> **Comment3: Some related works are missing.**
>
> A3: Thank you for pointing out these related works. We have added these related works in the revised version.

---

### Meta-Review · Area_Chair_g2Xa · 2022-09-15

**Recommendation:** Accept
**Confidence:** 5

**Metareview:**

This paper introduces the first NAS benchmarks on graphs. Ratings were quite diverse, with scores of 4,5,6,7,7,8.
The many positive reviewers highlighted that GraphNAS is very relevant and that a benchmark for them would be very useful for the community. The benchmark includes comprehensive evaluations on as many as 9 different datasets, which may also help facilitate research on meta-learning for (Graph)NAS. Initially, the code for creating the benchmark and analysis was not available, which is a no-go for tabular NAS benchmarks, but the authors fixed this in the rebuttal period.
The most negative reviewer, 8asK, appears to not be familiar with tabular NAS benchmarks and their usefulness, and did not react to the discussion about it, as well as to the modification of the paper to highlight it. The other borderline negative review, by reviewer LkTe, had as their main ciriticism that NAS methods should be benchmarked on many spaces, but the point of tabular NAS benchmarks is *not* to provide a conclusive assessment of the performance of NAS methods that holds in general across search spaces (as a "horserace" paper might try to achieve), but rather to facilitate the cheap evaluation of current and future NAS methods on a single search space. For these reasons, I do agree more with the positive reviewers and recommend acceptance of this work as a poster.

---

### Decision · Program_Chairs · 2022-09-16

Accept